# Rapid Transportation and Green Technology Innovation in Cities—From the View of the Industrial Collaborative Agglomeration

**Shanlang Lin** [†], **Ziyang Chen** [†] and **Ziwen He** [*,†]

School of Economics and Management, Tongji University, Tongji Building A, Siping Road 1500, Shanghai 200092, China; 05069@tongji.edu.cn (S.L.); 1911403@tongji.edu.cn (Z.C.)
* Correspondence: 1910450@tongji.edu.cn
† All authors contributed equally to this study and share the first authorship.

**Abstract:** This research uses a two-way fixed effect model to examine the correlation between the development of high-speed rail and the growth of green technology innovation by adopting the panel data of 284 Chinese cities between 2004 and 2013. The empirical results show that the supply of high-speed rail services has a significant promoting influence on the advance of urban green technology innovation capabilities. In particular, cities located in central and western China, along with those cities with relatively little government support or relatively backward public cultural infrastructure, have more chances to benefit from the supply of high-speed rail services. In addition, by employing a mediating effect model, this article finds that the industrial collaborative agglomeration plays an important mediating role between high-speed rail and regional green technology innovation. Therefore, this article suggests that the Chinese central government should continue to accelerate the construction of the rapid transportation network and expand the coverage of high-speed rail services in China to increase the growth of green technology innovation and achieve steady and sustained economic growth in China. Meanwhile, local governments should actively guide the collaborative agglomeration of manufacturing and related producer service industries under local conditions to stimulate the expansion of the green technology innovation market.

**Keywords:** high-speed rail; green technology innovation; industrial collaborative agglomeration; mediating effect





## 1. Introduction

Facing the dual constraints on resources and the environment, the demand for green technology innovation is gradually becoming crucial. The inconsistency between environmental protection and economic expansion is becoming one of the key factors restricting China and even the world from achieving sustainable development goals. It has become an urgent problem for all countries in the world to find an operative way to stimulate sustainable economic progress and economic green growth. Green technology innovation stimulates economic growth through traditional technology innovation. In the meantime, it saves resource consumption and reduces environmental pollution through new ideas and new technologies [1]. This is an effective means for a country to break through environmental and resource constraints and achieve green economic development. Hence, to acquire an advantage in the potential international competition, many countries regard the growth of green technology innovation as the guiding principle for their industrial development. Similarly, the Chinese government proposes to vigorously promote strategic emerging industries [2]. After years of exploration, China has made significant progress in green technology innovation. The "China Green Patent Statistics Report (2014–2017)" indicates that the quantity of green invention patent applications in 2017 increased by roughly 80% compared with 2014 [3]. However, based on the Environmental Performance

Index (EPI) released by Yale University in 2018, China's Environmental Performance Index ranks 120th among 180 regions [4]. This seems to indicate that China needs to develop green technology innovation better in the future to improve environmental efficiency and achieve sustainable economic development.

From 2003, when the "Leapfrog Development of China Railway" strategy was put forward, to 2021, when Chinese high-speed rail operating mileage became the largest in the world, reaching 36,000 km, the Chinese government has emphasized the significance of providing the people with a sounder and more modern railway system to satisfy their increasing demand for rapid transportation. The construction of Chinese high-speed rail, as an important engine of national modernization, has a wide influence on China's urban spatial layout, transportation mode, and economic development. Previous studies that show the relationship between the growth of Chinese high-speed rail and innovation indicate that the opening of high-speed rail accelerates the accessibility of different regions. Eventually, this result reduces the cost of commuting and accelerates the flow of talents, as well as innovation elements. It will stimulate the innovation output of enterprises and society and thus affect the innovation level of different regions. Striving to develop high-speed rail has become China's national strategy, and this will help the country to realize economic restructuring and promote regional green economic development. So, it is necessary for the Chinese Government to allocate social resources through high-speed rail to develop green technology innovation and advance green economic growth.

There has been little research on the explanation and identification of the influences of high-speed rail on regional green technology innovation to date. However, there is a consensus that the construction of high-speed rail raises the level of regional traditional innovation. Consequently, this article supplements the research content on the development of green technology innovation in China and examines the influences of high-speed rail construction on green technology innovation. Meanwhile, due to the trend of industrial integration, the Chinese government, learning from developed countries, is shifting its development model from a "manufacturing-driven" mode to a "two-wheel driven" mode, which is relying on the joint development of producer service industries and manufacturing. The government aims to boost regional economic growth and increase productivity while promoting regional green technology innovation. Therefore, this article introduces the collaborative agglomeration of manufacturing and producer services industries into an analysis for the first time and deeply analyzes the relationships between high-speed rail, industrial collaborative agglomeration, and green technology innovation.

The contributions and highlights of this article are mainly reflected in three aspects: firstly, this research proves that high-speed rail is a critical factor for promoting green technology innovation. This article not only analyzes the internal connection between the expansion of high-speed rail and regional green technology innovation in theory but also empirically tests this relationship by using the data of 284 Chinese cities between 2004 and 2013. Secondly, this paper establishes a theoretical framework that incorporates the collaborative agglomeration of manufacturing and producer services industries, high-speed rail, and green technology innovation for the first time. The intermediary role of industrial collaborative agglomeration is verified by the intermediary effect model. Thirdly, through the heterogeneity tests, it is found that the effects of green technology innovation brought about by high-speed rail are varied in different types of cities. Cities located in the western and central regions or that comparatively lack government support and are relatively backward in public cultural infrastructure are more able to obtain the benefits from the supply of high-speed rail services.

The remainder of the study is arranged as follows: Section 2 presents a review and summary of past related literature; Section 3 presents the theoretical analysis of high-speed rail, industrial collaborative agglomeration, and green technology innovation; Section 4 presents an empirical test of the relationship between the high-speed railway and green technology innovation and verifies that the industrial collaborative agglomeration is an intermediary variable between high-speed rail and green technology innovation; Section 5

reveals the empirical results and analysis; finally, Section 6 summarizes and contributes some helpful hints on policy.

## 2. Literature Review

Extensive research on green technology innovation and sustainable economic development has previously been conducted. As early as 1995, Porter and Porter [5] emphasized the role of technological innovation in sustainable development. Green technology innovation, as a part of traditional innovation, is the core element of helping society, organizations, and enterprises to gain competitive advantages and accomplish environmentally sustainable development [6–9]. Many scholars have stated that green technological innovation can not only seriously increase green total factor productivity [10] but also reduce the harmful impact on the environmental system [11,12]. Besides, some studies claim that green innovation activities related to environmental issues generate considerable knowledge spillovers, which are the key to achieving green economic growth [13]. Hence, promoting regional green technology innovation is becoming an inevitable choice for China to pursue sustainable economic growth under the conditions of limited environmental resources.

Innovation always plays a prominent part in the relationship between transportation development and economic benefits [14]. With the introduction of the green economy, it is particularly essential to study the correlation between high-speed rail and green technology innovation. Speeding up the flow of innovation elements between regions and promoting knowledge spillover have been considered as two important ways for high-speed rail to increase the effectiveness of innovation [12,15–18] because the accessibility of urban areas can be significantly improved by connecting to the to the high-speed railway. On the one hand, this will lead to a great reduction in transportation costs, which directly speeds up the flow of innovation elements between regions, resulting in regional knowledge diffusion and knowledge spillover. On the other hand, it eases the segmentation of the labor market and improves the effectiveness of searching and matching, thereby increasing the effectiveness of innovation [19]. Green technology innovation, as a specific type of innovation focusing on the environmental sector, will also be affected through the construction of high-speed rail. From the perspective of the development of innovation factors, Huang and Wang [20] found that high-speed rail can promote green technology innovation by producing technological progress effects and structural optimization effects. Although there is a relative lack of research directly related to green innovation, many scholars have demonstrated that high-speed rail is a key element affecting the ecological environment and green growth of a region [21–24]. Jia et al. [24] and Yang et al. [25] both pointed out that HSR can reduce environmental pollution through technological innovation. Green technology innovation, as a kind of innovation, is the key to reducing environmental pollution. Therefore, the authors of this article have reasons to believe that there is a link between high-speed rail and green technology innovation.

In the past, industrial agglomeration was generally presented in the form of the spatial agglomeration of a single industry. However, with the continuous rationalizing and upgrading of industrial structure, industrial agglomerations tend to involve collaborative agglomeration and mutual integration among heterogeneous industries, affecting urban economic production activities. In consequence, an industrial agglomeration between manufacturing and producer services has emerged. Many scholars claim that industrial collaborative agglomeration has the ability to accomplish the sustainable growth of green innovation [26–28] because it can bring about multiple effects, such as the industrial linkage effect, scale effect, and spillover effect, that play a valuable role in advancing the level of green innovation. In addition, the collaborative agglomeration of industries narrows the spatial distance of industrial integration, which is conducive to knowledge sharing and communication between industries [29–32]. As a result, the level of joint innovation and production efficiency will be improved, helping to continuously encourage the growth of green technology innovation.

Meanwhile, in the context of the rapid extension of high-speed rail, scholars have deeply explored the influence of high-speed rail on industrial agglomeration. As stated by traditional Location Theory and New Economic Geography, both the costs of transportation and allocation of resources play crucial parts in affecting industrial distribution [33,34]. On this basis, most empirical research illustrates how high-speed rail affects industrial agglomeration from the perspectives of market accessibility [35,36], knowledge spillover [37,38], and resource allocation [39]. Most researchers hold the view that there is a positive correlation between high-speed rail and industrial agglomerations in the three fields of manufacturing, high-tech industries, and producer services [40–42]. However, the authors of some studies do not believe that high-speed rail always has agglomeration effects, pointing out that the influences of high-speed rail on different regions and different economic sectors are varied [40,43].

The aforementioned literature offers a vital basis and ideas for this study, but there are many shortcomings. First of all, most of the existing literature only examines the internal connection between high-speed rail and traditional technological innovation, but research on high-speed rail and green technology innovation is lacking. Secondly, the relevant literature only explains the correlation between high-speed rail and green innovation efficiency from the perspective of the development of innovation factors, but further exploration of the internal mechanism of high-speed rail and green technology innovation is lacking. Additionally, researchers have not yet conducted an empirical evaluation of the role of industrial collaborative agglomeration in the correlation between high-speed rail and green technology innovation. Therefore, built on the hypotheses of New Economic Geography and agglomeration economy, it is reasonable to investigate the mediating role of industrial collaborative agglomeration between high-speed rail and green technology innovation.

## 3. Theoretical Analysis

Since the outbreak of the global environmental crisis, green technology innovation has become an important issue in academia and politics. To balance the development of society, economy, and environment, green technology innovation has been listed as the goal of the Chinese national innovation strategy. To be specific, boosting the growth of green technology innovation will help protect the ecological environment and achieve social and economic advancement at the same time. According to the literature review, it is learned that high-speed rail, because of its rapid passenger transport capability, can directly cause the rapid flow of human capital, improve the effectiveness of labor allocation, and bring about the diffusion of knowledge, which will directly promote green technology innovation. Furthermore, it is known that a high-speed rail service produces a considerable influence on industrial agglomeration, which is also an extensive factor affecting green technology innovation. Hence, this article further analyzes the mediating role of industrial collaborative agglomeration between high-speed rail and green technology innovation based on the hypotheses of New Economic Geography and agglomeration economics (as shown in Figure 1).

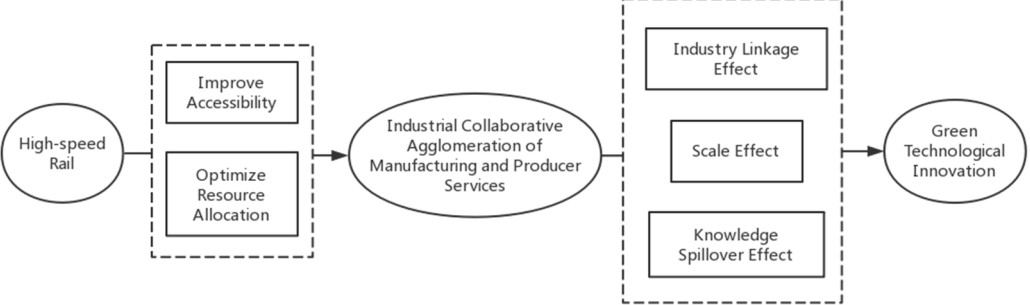

**Figure 1.** Mechanism diagram of high-speed rail, industrial collaborative agglomeration, and green technology innovation.

As stated by New Economic Geography and location theory, the spatial change of industrial distribution is a matter of the location selection of enterprises. High-speed rail, as a representative of rapid transportation services, is a principal element that affects the location of enterprises. High-speed rail improves the accessibility of a city, optimizes the allocation of resources, speeds up the information flow, and reduces transaction costs and manufacturing costs, attracting producer services and manufacturing and producing a longitudinal and transverse economic relation resulting in a collaborative agglomeration of these two major industries [44]. More specifically, high-speed rail reduces the cost of commuting. It can not only promote the transfer of the labor force between cities [45–48] but also induce the redistribution of productive resources such as capital information and technology between cities [16,39,49]. Generally, areas with intensive knowledge and production factors are preferred locations for enterprises. Through connecting to high-speed rail, regions gain location advantages such as fast information flow and optimized resource allocation. Simultaneously, the expansion of transportation caused by high-speed rail can also reduce trade costs [50]. The convenient transportation allows enterprises to locate in cities where labor costs or land prices are lower [51,52].

In addition, based on agglomeration economy, this article claims that industrial collaborative agglomeration can stimulate green technology innovation through industrial linkage effects, scale effects, and knowledge spillover effects. Firstly, the collaborative agglomeration of manufacturing and producer services shows a strong correlation between the upstream and downstream of the industry [53], which can enhance the effectiveness of resource allocation and utilization, improve the skill level of the labour force, and thus support the innovation of green technology. Industrial collaborative agglomeration means a clear industrial distribution of labor. The specialized segmentation of labor, which means that the manufacturing industry provides a large number of skilled professional labor forces while the producer service industry provides advanced and professional elements such as technology and information services as well as diversified financial resources [54], can increase the effectiveness of resource allocation. Besides, due to the sharing of professional inputs, such as labor, technology, information, and capital, the complementary advantages will be highlighted and the utilization rate of resources will be improved [55]. Secondly, collaborative agglomeration reduces corporate costs through the scale effect [56], which is beneficial to the creation of green technologies. Specifically, scale economies contribute to unify the factor market and establish a standard barrier of industry entry, which can help manufacturing and producer services industries to achieve the effective alignment of the value chain, thereby effectively reducing transaction costs [57]. Meanwhile, the collaborative agglomeration of manufacturing and producer services leads to a larger local market. For enterprises, there is an expansion of the local labor market and a decline in labor costs, which will help them save costs and expand profit margins [58]. Since capital constraints represent a serious hindrance to green technology innovation [59], cost savings and the expansion of profit margins will allow companies to have more funds to invest in green technology research, thereby improving corporate green technology innovative output. Thirdly, by means of the knowledge spillover effect, industrial collaborative agglomeration promotes green technology innovation. For manufacturing and producer services, the geographical proximity of the two sectors facilitates timely and frequent interaction and communication on issues related to the existing problems about energy consumption rates and environmental pollution efficiency [60]. In addition, due to the similar knowledge base and human capital, the cost of communication between the two departments will be reduced, which will facilitate the generation of new knowledge, thus continuously driving green technology innovation [61,62].

In summary, this article presents the following two basic assumptions:

**Hypothesis 1.** *The supply of high-speed rail services can considerably increase the level of a city's green technology innovation capabilities.*

**Hypothesis 2.** *High-speed rail stimulates the growth of urban green technology innovation by promoting the collaborative agglomeration of manufacturing and producer services.*

## 4. Key Indicators Measurement and Data Description

### 4.1. Green Technology Innovation

Currently, there are two core means of measuring green technology innovation: including input and output indicators at the same time, or only using a single output indicator. The input–output dual index model generally takes the productivity of both input and output as the proxy variable of green innovation evaluation with the consideration of the restraints in resources and environment. For example, Zeng et al. [57] divided the input variables into the input of research and development laborers, capital investment of research and development, and input of energy, and set the output variables as expected output and unexpected output. Then, they employed the improved super-SBM (Slack Based Measure) DEA (data envelopment analysis) model to measure the productivity of green innovation. However, Lu and Huang [63] pointed out that the inclusion of input indicators and output indicators at the same time would lead to the double counting of technological innovation measures. Regarding the method of using a single output indicator, which generally adopts the patent data, although there are some flaws such as ignoring the quality and economic impact of innovation, this approach has been considered reasonable by many scholars [64–67] because the change in the number of patents is determined by knowledge indicators such as investment in research and development and scientific research personnel, and patents the advantages of availability, measurability, and comparability in both time and space. In addition, it is noticeable that the examination process of invention patents is very strict, often taking 1 to 2 years to obtain authorization, and it is easily influenced by external issues such as the efficiency and preference of the patent licensing agency. Furthermore, the number of invention patent applications is more suitable to denote the current capabilities of regional green technology innovation. Hence, the quantity of green patent applications in prefecture-level cities per 10,000 people is finally used in this article as a proxy variable for green technology innovation. It is the sum of the number of patents, green invention patent applications, and green utility model patent applications.

### 4.2. The High-Speed Rail Service

Most studies on high-speed rail prefer to use the dummy variable of "whether high-speed rail is open or not" for empirical analysis [68–70], but this variable cannot be used to precisely distinguish the differences of the intensity and density of high-speed rail services provided between various cities. Therefore, in this article, we decided to measure the development level of high-speed rail in various regions by using the high-speed rail service frequency. Specifically, the software named "Superior Train Timetable" was used to count the total number of G-high-speed EMU trains and D-EMU trains that stop in each city in the final version of the regional train timetable at the end of each year to estimate the provided intensity of the high-speed rail service. Compared with dummy variables, which only have values of 0 and 1, the service frequency of high-speed rail can reflect the network density and actual operation of high-speed rail.

### 4.3. The Industrial Collaborative Agglomeration Index

To calculate the collaborative agglomeration index, it is essential to first assess the agglomeration index of manufacturing and producer services. There are several main indicators to assess the level of industrial agglomeration, including location entropy, the Gini coefficient, the Herfindahl–Hirschman Index, and employment density. Shao et al. [42] stated that the location entropy eradicates the differences in regional scale and additionally exposes the spatial allocation of geographic elements. Referring to their approach, this article employs location entropy to estimate the degree of specialized agglomeration of

manufacturing and producer service industries in prefecture-level cities. The equation is shown below:

$$\text{LocalEntropy}_{ir} = \frac{e_{ir}/e_i}{e_r/e}$$

where $e_{ir}$ indicates the output value or the number of employees of industry r in city i; $e_i$ represents the output value or the number of employees of all industries in the city i; $e_r$ is the output value or the number of employees of industry r in all cities; e means the output value or the number of employees of all industries in all cities. Generally, a large location entropy index means a high level of industrial agglomeration in an area. Due to the availability of data, for this paper, we finally decided to use the number of employees in manufacturing and eight types of productive service industries to calculate the degree of agglomeration. It should be noted that the eight types of productive service industries generally include "Traffic, Transport, Storage and Post", "Information Transmission, Computer Services and Software", "Financial Intermediation", "Real Estate", "Leasing and Business Services", "Scientific Research and Technical Service", "Services to Households and Other Services", and "Education" [28]. Then, according to the method of Zeng [57], the calculation of the industrial collaborative agglomeration index of manufacturing and producer service industries ($\text{COAGG}_{it}$) is as follows:

$$\text{COAGG}_{it} = \left(1 - \frac{|\text{MAGG}_{it} - \text{SAGG}_{it}|}{\text{MAGG}_{it} + \text{SAGG}_{it}}\right) + (\text{MAGG}_{it} + \text{SAGG}_{it})$$

where $\text{COAGG}_{it}$ is the industrial collaborative agglomeration index of manufacturing and producer services in city i in year t; $\text{MAGG}_{it}$ describes the manufacturing agglomeration index of city i in year t; and $\text{SAGG}_{it}$ denotes the agglomeration index of the producer service industry in city i in year t. Specifically, $\left(1 - \frac{|\text{MAGG}_{it} - \text{SAGG}_{it}|}{\text{MAGG}_{it} + \text{SAGG}_{it}}\right)$ stands for the quality of the collaborative agglomeration index while $(\text{MAGG}_{it} + \text{SAGG}_{it})$ means the depth. A large industrial collaborative agglomeration index signifies that the quality of collaborative agglomeration is great and the depth is deep.

### 4.4. Control Variables

With the purpose of controlling other features that may lead to differences in urban green technology innovation, this paper adds six covariates. (1) Government support (sci_GDP) is characterized by the proportion of fiscal expenditure on science in GDP. The government's financial support is vital to green technology innovation [71]. (2) Financial development (Fin_GDP) is represented by the proportion of the total amount of deposits to and loans from financial institutions in GDP at the end of the year. The financial system can provide services such as collecting and analyzing information, diversifying risk, and rationalizing credit, which will have an impact on the continuation of green technology innovation activities and the enhancement of green technology innovation efficiency [72]. (3) The public cultural infrastructure (lnbooks_100_1) is valued by the logarithm of the number of books collected per 100 people. Cities with a high level of public cultural infrastructure are more likely to attract innovative talents, cause knowledge spillover, and develop the growth of green technology innovation [73]. (4) The level of communication technology (lnmobile1) is estimated by the logarithm of the number of mobile users at the end of the year. Communication technology enhances the convenience of knowledge exchange and promotes long-distance knowledge spillover, which will have an essential influence on the dissemination of knowledge, thereby affecting the level of regional green technology innovation. (5) The vitality of the economic system (umemp_Pop) is characterized by the ratio of the unemployed population to the total population at the end of the year because a long-term increase in the unemployment rate indicates that the economy is stagnant and investment capacity is reduced, which may lead to a decline in the rate of technology innovation, leading to low knowledge production capacity, resulting in low knowledge production capacity, and inhibiting green

technology innovation. (6) The degree of environmental pollution (gyfs_GDP) is measured by industrial effluent emissions per unit of GDP. The higher the pollutant emissions per unit of GDP, the worse the environmental quality, which will restrict the development of the local economy to a certain extent, thus inhibiting technological innovation.

*4.5. Data Source*

The main parts of the sample data were derived from the China Urban Statistical Yearbook and China Regional Economic Statistical Yearbook; green patent data were obtained from the State Intellectual Property Office; high-speed rail-related data were taken from the Superior Train Timetable software. In this article, we mainly focus on the green technology innovation performance of Chinese prefecture-level cities from 2004 to 2013 for a total of 10 years. Considering the lack of statistical data in some cities, for this article, we finally chose 284 cities as the research sample. Additionally, to reduce the heteroscedasticity in the empirical analysis, for this article, we adopted the logarithm of some variables and added 1 to variables with values of zero before taking the logarithm (see Table 1).

**Table 1.** Statistical description of variables.

| Variable Name | Number of Observation | Mean | Standard Deviation | Min | Max |
|---|---|---|---|---|---|
| Green technology innovation (lnGreenPatent) | 2840 | −2.450055 | 1.583217 | −6.310173 | 2.618316 |
| High-speed rail service (lnHSRFreq1) | 2840 | 0.7096632 | 1.469769 | 0 | 6.381816 |
| Government support (sci_GDP) | 2840 | 0.1438235 | 0.1501863 | 0 | 1.996286 |
| Financial development (Fin_GDP) | 2840 | 3.360883 | 0.9064529 | 0 | 8.372153 |
| Public cultural infrastructure (ln books_100_1) | 2840 | 5.085405 | 1.217877 | −9.21034 | 8.123795 |
| Level of communication technology (ln mobile1) | 2840 | 1.9142 | 0.9666132 | 0 | 10.74006 |
| Vitality of the economic system (umemp_Pop) | 2840 | 0.6173442 | 0.5465437 | 0 | 11.54366 |
| Degree of environmental pollution (gyfs_GDP) | 2840 | 0.9414255 | 1.298091 | 0 | 28.92866 |

## 5. Empirical Model

*5.1. Baseline Model*

To confirm the influence of a high-speed rail service supply on urban green technology innovation capability, we developed the following econometric model:

$$\ln \mathrm{GreenPatent}_{i,t} = \alpha_0 + \alpha_1 \ln \mathrm{HSRFreq1}_{i,t} + \alpha_2 X_{i,t} + \mu_i + \gamma_t + \varepsilon_{i,t}$$

where i and t denote city and year, respectively; $\ln \mathrm{GreenPatent}_{i,t}$ is the explained variable representing the logarithm of the number of green patents of city i in year t; $\ln \mathrm{HSRFreq1}_{i,t}$ is the main explanatory variable, standing for the frequency of high-speed rail service in city i in year t; $X_{i,t}$ is a series of control variables, involving government support, financial development, the public cultural infrastructure, the level of communication technology, the vitality of the economic system, and the degree of environmental pollution; $\mu_i$ indicates the city fixed effect, controlling the city characteristic factors which do not change with time; $\gamma_t$ denotes the year fixed effect, which is employed to control the urban heterogeneity; and $\varepsilon_{i,t}$ is the random error term.

*5.2. Mediating Effect Model*

The previous theoretical analysis shows that high-speed rail is able to enhance the degree of industrial coordination and agglomeration, which in turn makes a difference regarding the level of green technology innovation. Thereby, this article refers to the classic three-step model [74] to construct an intermediary effect model to test the intermediary effect of industrial collaborative agglomeration. The specific steps are as follows:

Step 1: Assess the effect of high-speed rail on green patents (baseline model);

$$\ln \text{GreenPatent}_{i,t} = \alpha_0 + \alpha_1 \ln \text{HSRFreq1}_{i,t} + \alpha_2 X_{i,t} + \mu_i + \gamma_t + \varepsilon_{i,t}$$

Step 2: Test the influence of high-speed rail on industrial collaborative agglomeration;

$$\text{COAGG}_{i,t} = \beta_0 + \beta_1 \ln \text{HSRFreq1}_{i,t} + \beta_2 X_{i,t} + \mu_i + \gamma_t + \varepsilon_{i,t}$$

Step 3: Introduce the Industrial collaborative agglomeration into baseline model.

$$\ln \text{GreenPatent}_{i,t} = \gamma_0 + \gamma_1 \ln \text{HSRFreq1}_{i,t} + \gamma_2 \text{COAGG}_{i,t} + \gamma_3 X_{i,t} + \mu_i + \gamma_t + \varepsilon_{i,t}$$

To test the mediation effect, it is first necessary to pass the baseline model regression test. Then, a further test is needed to prove that the influence of high-speed rail on the industrial collaborative agglomeration is significant. Finally, the collaborative agglomeration is introduced into the baseline model to form a new empirical model, and there is a need to observe the changes in regression coefficients. If the coefficients $\alpha_1$, $\beta_1$, $\gamma_1$, and $\gamma_2$ are all significant, and $\alpha_1 > \gamma_1$ or the significance of $\gamma_1$ is lower than the significance of $\alpha_1$, then it is believed that the mediation effect exists, which means that high-speed rail can influence the advance of green technology innovation by involving industrial collaborative aggregation. Otherwise, the transmission mechanism does not exist, and the analysis is ended.

## 6. Empirical Results and Analysis

Before the regression analysis of all models, a multicollinearity test was performed on panel data. The results showed that the maximum variance inflation factor (VIF) was 1.55 (as shown in Table 2) and therefore less than 10. So, it was not necessary to consider multicollinearity. In addition, the test of individual effects suggested that there were individual effects. Hence, the mixed regression should not be used. Besides, we conducted an overidentification test to determine that the fixed effects model, rather than the random effects model, should be used in this article.

**Table 2.** Multicollinearity test.

| Variables | VIF | 1/VIF |
|---|---|---|
| lnbooks_100_1 | 1.55 | 0.645504 |
| sci_GDP | 1.49 | 0.671654 |
| lnHSRFreq1 | 1.45 | 0.691791 |
| lnmobile1 | 1.42 | 0.703521 |
| Fin_GDP | 1.37 | 0.731019 |
| unemp_Pop | 1.14 | 0.873686 |
| gyfs_GDP | 1.07 | 0.932161 |
| Mean VIF | 1.36 | |

*6.1. Results of Baseline Model Regression*

Under the premise of using a two-way fixed effect model, we tested each variable individually to ensure the robustness of the regression results. The specific regression results are displayed in Table 3. All estimation results revealed that, after adding the control variables individually, the influence of the high-speed rail service frequency on urban green technology innovation always remained dramatically optimistic at the confidence level of

1%, and there was little difference in magnitude. As shown by the coefficient in column (7), for every 1% increase in the frequency of high-speed rail services in a city, the number of green patent applications per 10,000 people increases significantly by about 9.1%. This implies that high-speed rail is a critical catalyst for urban green technology innovation improvement and is able to encourage the advance of green technology innovation effectively.

**Table 3.** The empirical results of the baseline model.

|  | (1) | (2) | (3) | (4) | (5) | (6) | (7) |
|---|---|---|---|---|---|---|---|
| **Variables** | | | | | | | |
| lnHSRFreq1 | 0.101 *** | 0.090 *** | 0.088 *** | 0.086 *** | 0.087 *** | 0.088 *** | 0.091 *** |
| | (4.80) | (4.13) | (4.14) | (4.06) | (4.08) | (4.09) | (4.21) |
| sci_GDP | | 0.556 * | 0.506 * | 0.548 * | 0.588 * | 0.594 * | 0.581 * |
| | | (1.86) | (1.68) | (1.80) | (1.92) | (1.94) | (1.91) |
| lnbooks_100_1 | | | 0.105 * | 0.139 *** | 0.145 *** | 0.144 *** | 0.144 *** |
| | | | (1.90) | (2.98) | (3.13) | (3.12) | (3.12) |
| lnmobile1 | | | | −0.093 ** | −0.088 ** | −0.087 ** | −0.086 ** |
| | | | | (−2.53) | (−2.46) | (−2.43) | (−2.41) |
| Fin_GDP | | | | | −0.131 ** | −0.126 ** | −0.121 ** |
| | | | | | (−2.46) | (−2.40) | (−2.33) |
| unemp_Pop | | | | | | −0.086 ** | −0.085 ** |
| | | | | | | (−2.30) | (−2.23) |
| gyfs_GDP | | | | | | | −0.068 *** |
| | | | | | | | (−2.60) |
| Constant | −3.368 *** | −3.385 *** | −3.713 *** | −3.434 *** | −3.220 *** | −3.178 *** | −3.088 *** |
| | (−48.43) | (−47.59) | (−20.68) | (−16.02) | (−13.82) | (−13.48) | (−12.73) |
| Observations | 2840 | 2840 | 2840 | 2840 | 2840 | 2840 | 2840 |
| R-squared | 0.348 | 0.350 | 0.351 | 0.355 | 0.357 | 0.358 | 0.360 |
| Number of cities | 284 | 284 | 284 | 284 | 284 | 284 | 284 |
| City FE | Yes | Yes | Yes | Yes | Yes | Yes | Yes |
| Time FE | Yes | Yes | Yes | Yes | Yes | Yes | Yes |

Robust t-statistics in parentheses. *** $p < 0.01$, ** $p < 0.05$, * $p < 0.1$.

By viewing the coefficients of the control variables, it can be found that government support and public cultural infrastructure have a meaningful positive effect on the output of green technology innovation. To some extent, government support can compensate for the higher research and development costs and external risks in the innovation activities of green innovative enterprises. Meanwhile, the per capital book collection in a city can reflect a city's basic amount of knowledge reserves. This shows that a city with a stronger public cultural foundation has a stronger ability to innovate in green technology. Conversely, the regression results show that the level of communication technology, financial development, economic system vitality, and the degree of environmental pollution have a considerable negative correlation with the number of green patent applications per 10,000 people. This can be clarified by the fact that, although the improvement of information and communication technology has improved the convenience of people's communication, it has reduced the demand for face-to-face communication among technicians, thus inhibiting the spread of knowledge related to green technology to a certain extent. Besides, despite the high level of financial development, funds may not flow to the green technology innovation market with high-risk characteristics, leading to the problem of financing constraints for enterprises in green innovation activities. Next, the increase in the unemployed population will cause local governments to shift their development focus from attracting high-tech personnel and developing green innovations to improving people's employment and welfare. Furthermore, cities with high levels of environmental pollution are generally those with secondary industries such as manufacturing and industry as the focus of local economic development; cities located in central and western China are also particularly affected by pollution. These cities usually have unclear goals for sustainable economic development

and low environmental protection requirements, which work against the development of green technology creation.

### 6.2. Results of Mediating Effect Model Regression

According to the setting of the mediating effect model constructed by the classic three-step method, the collaborative agglomeration of manufacturing and producer services is introduced to test the relationship between high-speed railway, industrial collaborative agglomeration, and green technology innovation. Table 4 reports the regression results of the mediation effect model. From the results of columns (1) to (3), the coefficient of the high-speed rail service frequency is always positive and exceeds the significance test of 5%. Column (1) demonstrates the results of baseline regression, while column (2) shows that the supply of high-speed rail services significantly enhances the degree of urban industrial collaborative agglomeration. It is clear that when the frequency of urban high-speed rail services increases by 1%, the city's industrial collaborative aggregation increases by 4.8%. Additionally, it can be found that there is a gap in the coefficient of the high-speed rail service frequency between column (1) and column (3), which is reduced from 0.091 to 0.089. Meanwhile, by observing the results of column (3), the coefficient of collaborative agglomeration for manufacturing and producer services is considerably positive at the confidence level of 5%, which is 0.046. This indicates that there is a remarkable and optimistic relationship between industrial collaborative agglomeration and green technology innovation. Hence, in this article, we can determine that industrial collaborative agglomeration plays a mediating role, and the facilitation effect of high-speed rail on green technology innovation can be realized by promoting the industrial collaborative agglomeration of manufacturing and producer services.

**Table 4.** The empirical results of the mediating effect model.

|  | (1) | (2) | (3) |
| --- | --- | --- | --- |
| **Variables** | **lnGreenPatent** | **COAGG_ps8Manu** | **lnGreenPatent** |
| lnHSRFreq1 | 0.091 *** | 0.048 ** | 0.089 *** |
|  | (4.21) | (2.23) | (4.11) |
| COAGG_ps8Manu |  |  | 0.046 ** |
|  |  |  | (2.13) |
| sci_GDP | 0.581 * | 0.104 | 0.576 * |
|  | (1.91) | (0.36) | (1.91) |
| lnbooks_100_1 | 0.144 *** | 0.027 | 0.143 *** |
|  | (3.12) | (0.72) | (3.12) |
| lnmobile1 | −0.086 ** | −0.009 | −0.085 ** |
|  | (−2.41) | (−0.62) | (−2.40) |
| Fin_GDP | −0.121 ** | 0.062 | −0.124 ** |
|  | (−2.33) | (1.55) | (−2.38) |
| unemp_Pop | −0.085 ** | 0.045 * | −0.087 ** |
|  | (−2.23) | (1.78) | (−2.27) |
| gyfs_GDP | −0.068 *** | 0.000 | −0.068 *** |
|  | (−2.60) | (0.03) | (−2.60) |
| Constant | −3.088 *** | 2.582 *** | −3.206 *** |
|  | (−12.73) | (19.70) | (−13.27) |
| Observations | 2840 | 2840 | 2840 |
| R-squared | 0.360 | 0.016 | 0.361 |
| Number of cities | 284 | 284 | 284 |
| City FE | Yes | Yes | Yes |
| Time FE | Yes | Yes | Yes |

Robust t-statistics in parentheses. *** $p < 0.01$, ** $p < 0.05$, * $p < 0.1$.

### 6.3. Heterogeneity Test

There are obvious regional differences in the level of green technology innovation in China, especially regarding the eastern cities, which are developing rapidly at a level higher

than the national average. So, it is necessary for this paper to further study whether there are considerable regional differences in the promoting force of high-speed rail. Specifically, this article divides the sample into eastern, central, and western cities on the basis of prefecture-level cities to separately explore the effect of high-speed rail service frequency. The results of sub-sample regression are displayed in Table 5. Column (1) lists the regression results of 115 cities located in eastern China. The regression coefficient is positive, but it is not statistically significant. This means that for eastern cities, high-speed rail is unable to stimulate the progress of their green technology innovation level effectively. Huang and Wang [20] also observed similar results. Combined with their analysis, this article claims that there are two major explanations for this result. Firstly, the level of green innovation in most parts of the east has been relatively developed, which means that eastern cities have no additional room for significant growth in the short term. Secondly, a high-speed rail connection will cause more people to flock to developed cities that are already overloaded. This will encourage the government to fund the supply of basic social public services rather than green innovation, thereby constraining the city's green technology innovation.

**Table 5.** The results of the sub-sample regression of eastern, central, and western regions.

| | (1) | (2) | (3) |
|---|---|---|---|
| **Variables** | **Eastern China** | **Central China** | **Western China** |
| lnHSRFreq1 | 0.039 | 0.057 ** | 0.332 *** |
| | (1.64) | (2.10) | (3.80) |
| sci_GDP | 0.247 | 1.175 ** | −2.174 |
| | (0.58) | (2.51) | (−1.43) |
| lnbooks_100_1 | 0.196 *** | 0.082 | 0.063 |
| | (3.76) | (1.59) | (0.48) |
| lnmobile1 | −0.021 | −0.164 *** | −0.680 ** |
| | (−1.27) | (−4.37) | (−2.18) |
| Fin_GDP | 0.039 | −0.024 | −0.050 |
| | (0.88) | (−0.49) | (−0.40) |
| unemp_Pop | 0.025 | −0.029 | −0.539 ** |
| | (0.73) | (−0.54) | (−2.55) |
| gyfs_GDP | −0.041 | −0.050 | −0.217 |
| | (−1.06) | (−0.99) | (−1.49) |
| Constant | −3.459 *** | −3.343 *** | −0.276 |
| | (−15.62) | (−13.49) | (−0.20) |
| Observations | 1150 | 1080 | 600 |
| R-squared | 0.542 | 0.531 | 0.168 |
| Number of cities | 115 | 108 | 60 |
| City FE | Yes | Yes | Yes |
| Time FE | Yes | Yes | Yes |

Robust t-statistics in parentheses. *** $p < 0.01$, ** $p < 0.05$, * $p < 0.1$.

Column (2) reports the regression results of 108 cities in central China. It can be seen that the coefficient of high-speed rail service frequency is 0.057, and it is significantly positive at the 5% confidence level. This indicates that for the central region, the supply of high-speed rail services is an important factor in enhancing the level of green technology innovation. The governments of central cities have always been focused on providing a better employment environment to attract talented and high-skilled workers to stimulate their development. The transportation convenience brought by high-speed rail construction accelerates the transfer of some high-tech workers to middle and small-sized cities in the central region, which has a positive impact on the level of the city's green technology innovation.

From the regression result of column (3), it is proved that high-speed rail has brought about a dramatic and positive influence on green technology innovation in 60 western cities. This may be because the green technology innovation level of western cities is relatively backward compared with that of central and eastern cities, and the gap is relatively large.

However, the supply of high-speed rail services has greatly reduced transportation and commuting costs, sped up the development of talents and innovative elements, accelerated the dissemination of various knowledge and information, and formed a green innovation network between cities. As a result, the innovation output of society and enterprises is accelerated, and thus the level of green technology innovation in various western cities is greatly increasing.

To further test the heterogeneity of the effect of high-speed rail green technology innovation, this article adds the interaction terms of the high-speed rail service frequency and various control variables based on the baseline model regression. The results are displayed in Table 6. By observing columns (1) to (6), it can be found that only the coefficients of the interaction term between the high-speed railway and government support and public cultural infrastructure are significantly negative, while coefficients of other interaction terms are not statistically significant. This shows that the greater the degree of government support and the higher the level of public cultural infrastructure in a city, the smaller the marginal effect that the city will gain from the construction of high-speed rail. Meanwhile, this implies that cities that lack government support and that have relatively backward public cultural infrastructure can gain more benefits from the supply of high-speed rail services.

**Table 6.** The result of the further test of heterogeneity.

| | (1) | (2) | (3) | (4) | (5) | (6) |
|---|---|---|---|---|---|---|
| **Variables** | | | | | | |
| lnHSRFreq1 | 0.147 *** | 0.205 *** | 0.021 | 0.118 *** | 0.104 *** | 0.073 *** |
| | (5.92) | (3.85) | (0.14) | (3.40) | (3.97) | (2.80) |
| sci_GDP | 1.146 *** | 0.707 ** | 0.554 * | 0.633 ** | 0.602 * | 0.597 ** |
| | (3.15) | (2.26) | (1.72) | (2.03) | (1.94) | (1.99) |
| lnbooks_100_1 | 0.136 *** | 0.170 *** | 0.146 *** | 0.143 *** | 0.142 *** | 0.144 *** |
| | (2.92) | (3.49) | (3.16) | (3.09) | (3.07) | (3.11) |
| lnmobile1 | −0.086 ** | −0.087 ** | −0.086 ** | −0.086 ** | −0.086 ** | −0.085 ** |
| | (−2.39) | (−2.41) | (−2.41) | (−2.42) | (−2.41) | (−2.40) |
| Fin_GDP | −0.128 ** | −0.120 ** | −0.122 ** | −0.116 ** | −0.121 ** | −0.122 ** |
| | (−2.45) | (−2.32) | (−2.34) | (−2.20) | (−2.33) | (−2.34) |
| unemp_Pop | −0.090 ** | −0.087 ** | −0.086 ** | −0.085 ** | −0.077 ** | −0.085 ** |
| | (−2.43) | (−2.31) | (−2.23) | (−2.24) | (−1.97) | (−2.25) |
| gyfs_GDP | −0.065 ** | −0.067 ** | −0.068 *** | −0.067 ** | −0.067 ** | −0.066 ** |
| | (−2.52) | (−2.56) | (−2.62) | (−2.55) | (−2.58) | (−2.54) |
| lnHSRFreq1_sci_GDP | −0.213 *** | | | | | |
| | (−3.47) | | | | | |
| lnHSRFreq1_lnbooks_100_1 | | −0.029 ** | | | | |
| | | (−2.37) | | | | |
| lnHSRFreq1_lnmobile1 | | | 0.011 | | | |
| | | | (0.48) | | | |
| lnHSRFreq1_Fin_GDP | | | | −0.011 | | |
| | | | | (−0.96) | | |
| lnHSRFreq1_unemp_Pop | | | | | −0.017 | |
| | | | | | (−0.87) | |
| lnHSRFreq1_gyfs_GDP | | | | | | 0.039 |
| | | | | | | (1.61) |
| Constant | −3.063 *** | −3.165 *** | −3.088 *** | −3.092 *** | −3.086 *** | −3.088 *** |
| | (−12.54) | (−12.66) | (−12.73) | (−12.72) | (−12.73) | (−12.74) |
| Observations | 2840 | 2840 | 2840 | 2840 | 2840 | 2840 |
| R-squared | 0.362 | 0.361 | 0.360 | 0.360 | 0.360 | 0.361 |
| Number of the cities | 284 | 284 | 284 | 284 | 284 | 284 |
| City FE | Yes | Yes | Yes | Yes | Yes | Yes |
| Time FE | Yes | Yes | Yes | Yes | Yes | Yes |

Robust t-statistics in parentheses. *** $p < 0.01$, ** $p < 0.05$, * $p < 0.1$.

### 6.4. Robustness Tests

To validate the validity of the results, for this paper, we employed three methods for robustness tests. Firstly, considering the potential bias caused by the existence of outliers in the sample, the number of samples was reduced from 284 to 264, excluding the top 20 cities with the largest average number of green patents. The detailed regression results appear in column (1) of Table 7. Next, column (2) displays the result of the method of replacing the main explanatory variables, in which the logarithm of high-speed rail service frequency was replaced by dummy variables for the opening of high-speed rail. Finally, in consideration of the hysteresis effect of high-speed rail on the number of green invention patents granted, the variable of high-speed rail service frequency was treated with a lag of one period. The result is presented in column (3). The coefficients of high-speed rail always remained notably positive, showing that high-speed rail considerably effects the growth of green technology innovation in cities. Hence, it can be said that the core results of this paper are reliable and robust.

**Table 7.** The results of the robustness test.

| | (1) | (2) | (3) |
|---|---|---|---|
| **Variables** | **Exclude Outlier** | **dumHSR** | **L_lnHSRFreq1** |
| lnHSRFreq1 | 0.093 *** | | |
| | (3.89) | | |
| sci_GDP | 0.771 * | 0.871 ** | 0.555 ** |
| | (1.96) | (2.32) | (2.19) |
| lnbooks_100_1 | 0.142 *** | 0.144 *** | 0.118 *** |
| | (2.85) | (3.08) | (2.59) |
| lnmobile1 | −0.084 ** | −0.086 ** | −0.054 * |
| | (−2.16) | (−2.43) | (−1.81) |
| Fin_GDP | −0.133 ** | −0.119 ** | −0.123 ** |
| | (−2.33) | (−2.32) | (−2.18) |
| unemp_Pop | −0.101 ** | −0.084 ** | −0.086 * |
| | (−2.34) | (−2.29) | (−1.76) |
| gyfs_GDP | −0.066 ** | −0.060 ** | −0.047 * |
| | (−2.49) | (−2.32) | (−1.80) |
| dumHSR | | 0.356 *** | |
| | | (5.36) | |
| L_lnHSRFreq1 | | | 0.084 *** |
| | | | (4.05) |
| Constant | −3.234 *** | −3.106 *** | −2.961 *** |
| | (−12.70) | (−12.89) | (−12.08) |
| Observations | 2640 | 2840 | 2556 |
| R-squared | 0.335 | 0.361 | 0.346 |
| Number of the cities | 264 | 284 | 284 |
| City FE | Yes | Yes | Yes |
| Time FE | Yes | Yes | Yes |

Robust t-statistics in parentheses. *** $p < 0.01$, ** $p < 0.05$, * $p < 0.1$.

### 6.5. Endogenous Problems

In this paper, we controlled a series of features that affect the number of urban green patent applications per 10,000 people in the baseline model regression. Therefore, it is considered that the location choice of a high-speed rail connection is unlikely to depend on the spatial pattern of green innovation output. However, some endogenous variables that affect local green technology innovation may still be omitted in the process of setting a model, leading to deviations in the estimation results. It is necessary for the instrumental variables to meet the conditions of correlation and exogeneity; thus, for this article, we applied the average altitude in urban geographic information to construct instrumental variables for high-speed rail by referring to the method used by Yang et al. [25]. There is a negative relationship between altitude and high-speed rail construction, and altitude is an

exogenous geographic information variable that will not directly affect the level of green technology innovation in a city. Moreover, the average altitude generally does not change in a short period. As a result, after comprehensively considering the above factors, we finally selected the data of the last year of the sample (the year of 2013) for the regression of instrumental variables. The specific regression results are presented in Table 8. Column (1) reports the first-stage regression result of high-speed rail service frequency to the average altitude, confirming that they are negatively correlated. From column (2), it is observed that the regression results of the instrumental variable match the baseline model regression results; that is, the supply of high-speed rail services appreciably improves a city's green technology innovation level. In addition, it is observed that the F statistic value is 24.51, which is greater than 10; this indicates that the hypothesis of weak instrumental variables is rejected, proving that the instrumental variable is effective.

**Table 8.** The results of the regression using the instrumental variable.

|  | (1) | (2) |
| --- | --- | --- |
| **Variables** | **FirstStage** | **2SLS** |
| lnAltitute | −0.265 *** | |
|  | (−3.04) | |
| lnHSRFreq1 | | 0.833 *** |
|  | | (2.80) |
| Observations | 284 | 284 |
| R-squared | | −0.207 |
| Control Variables | Yes | Yes |
| F | | 24.51 |

Robust t-statistics in parentheses. *** $p < 0.01$, ** $p < 0.05$, * $p < 0.1$.

## 7. Conclusions and Implications

Understanding the relationship between high-speed rail and the level of green technology innovation is critical to investigating how transportation infrastructure will shape the future development of Chinese green technology innovation. In this paper, we applied panel data from 284 prefecture-level and above cities in China between 2004 and 2013 to construct a two-way fixed-effect model to evaluate the influence of high-speed rail service supply on green technology innovation. The regression results suggest that the supply of high-speed rail services can considerably promote the capability of urban green technology innovation. In addition, we creatively introduced the collaborative agglomeration of manufacturing and producer services to clarify the intermediary effect of the industrial collaborative agglomeration between high-speed rail and green technology innovation through the mediating effect model. Furthermore, from the heterogeneity tests, we found that the effects of green technology innovation brought about by high-speed rail are varied in different types of cities. Therefore, this article proves that high-speed rail considerably advances the level of green technology innovation through industrial collaborative agglomeration.

The basic results are consistent with the previous findings that high-speed rail speeds up the flow and diffusion of human capital and innovation factors between cities, thereby stimulating the development of urban green technology innovation. Meanwhile, by causing the collaborative agglomeration of manufacturing and producer services, high-speed rail expedites the formation of green technology innovation from three aspects: the industrial linkage effect, scale effect, and knowledge spillover effect. However, the results suggest that the effect of green technology innovation brought about by high-speed rail is not significant in eastern China. The possible reason is that most cities in the east are cities with advanced green technology innovation, and there is no room for additional growth in the short term. For cities in the central and western regions, high-speed rail is an essential catalyst for encouraging the growth of urban green technology innovation. Besides, after the further analysis of heterogeneity, we propose that cities that relatively lack government

support and that are relatively backward in public cultural infrastructure are more able to benefit in terms of their green technology innovation from the supply of high-speed rail services.

This study supplements the content related to the advance of green technology innovation, but there are some limitations. Firstly, this article only conducts empirical tests from the perspective of industrial collaborative agglomeration, leading to a lack of a comprehensive and detailed investigation on the mechanism of high-speed railways. Secondly, the spatial spillover effect of the high-speed railway is not taken into account, which may lead to a deviation in the estimation results. Finally, due to the availability of data, this article fails to give a more detailed classification of green patents. In the future, the relationship between the transportation network and various green patents will be discussed in a more comprehensive and in-depth way.

According to the main research conclusions, we propose the following recommendations by combining the status quo and characteristics of China's development. Firstly, China should continue to vigorously develop rapid transportation, expanding the coverage of high-speed rail services within China. In particular, the Chinese government should pay attention to the construction of high-speed rail in the central and western regions, which have less developed economies and inconvenient transportation. This is because a qualitative leap in transportation can significantly enhance the ability of these regions to innovate green technologies and improve the sustainability of their economies. Secondly, as China's economic development enters a new normal, the green upgrading and optimization of the industrial structure are the internal driving forces that will retain the steady and sustained growth of China's economy. The industrial collaborative agglomeration of manufacturing and producer services, as the two factors driving China's development, can reduce the cost of intermediate inputs in the manufacturing industry and give full play to the knowledge sharing and the influence of the service industry, thus forming economies of scale in the agglomeration areas. This will optimize the regional industrial structure, increase the level of regional green technology innovation, and stimulate the green growth of the Chinese economy. As a result, the Chinese government should improve the construction of transportation infrastructure, guide capital investment, and strengthen local public cultural infrastructure to effectively accelerate the collaborative agglomeration of manufacturing and producer services and increase the level of urban green technology innovation. Moreover, given the obvious regional differences in China, each local government should take the characteristics and conditions of the local manufacturing industry into consideration, and then actively guide the development of related producer services, thereby promoting the positive interaction and collaborative positioning between these two industries. Meanwhile, appropriate incentives and preferential policies can stimulate the expansion of the green technology innovation market.

**Author Contributions:** Conceptualization, Z.C. and Z.H.; methodology, Z.C.; software, Z.H.; validation, S.L., Z.C. and Z.H.; formal analysis, Z.C.; investigation, Z.C.; resources, Z.H.; data curation, Z.H.; writing—original draft preparation, Z.C.; writing—review and editing, Z.H.; visualization, Z.C.; supervision, S.L.; project administration, S.L.; funding acquisition, None. All authors have read and agreed to the published version of the manuscript.

**Funding:** Financial support was not provided for the performance of this research; this is a self-funded work.

**Institutional Review Board Statement:** Not applicable.

**Informed Consent Statement:** Not applicable.

**Data Availability Statement:** The data that support the findings of this study are available on request from the corresponding author. The data are not publicly available due to privacy or ethical restrictions.

**Conflicts of Interest:** The authors declare that they have no competing financial interest.

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
