# Peer review of "Rapid Transportation and Green Technology Innovation in Cities—From the View of the Industrial Collaborative Agglomeration"

_applsci, doi:10.3390/app11178110_

Round 1

Reviewer 1 Report

After reading the paper I have still doubts that a relationship exists between green technology innovation and high-speed rail connections. Although I accept the idea that there exists a relationship between the level of high-speed rail connections, people and businesses agglomeration and development that may be more or less linked to innovation, it is not easy to accept the idea that the level of high-speed rail connection is related to the specific kind of green innovation. More sound arguments are necessary to support such an assumption. Thus, my suggestion to authors is to investigate also the relationship considering all kind of innovation and not only the green technology innovation.

The authors should better justify the choice of variables they use in their model. A more in-depth literature review would help achieving this goal.

Did authors test also a random effect regression model? Why did they choose a fixed effect model?

Is there any lag between the dependent and independent variables? As I understand, authors assume that the measurements of independent variables at time t affect the measurement of dependent variable at time t. Is that true? Why? Are there any non-linear effects? Time has relevance because of the delay of learning and selection effects.

Please, discuss limitations of the study.

Reviewer 2 Report

This reviewer would like to thank the authors for this manuscript.

The manuscript addresses topics of very high importance (rapid transportation and green technology), and it has novelty.

While the manuscript has merit for publication, this reviewer would like to share the following two questions, which mainly refer to the structure and style of the manuscript:

Question 1. Unless otherwise advise by the journals’ guidelines: Could sections “2. Literature review” and “3. Theoretical analysis” be shortened and eventually merged?

The current manuscript structure resembles nearly too much that of a degree thesis document.

Question 2: While results are discussed in the present manuscript version: Would the authors consider that the manuscript would exhibit its inherent strength clearer, and would become more reader friendly, if the manuscript would have a discussion section more clearly identified? Could the authors consider this, and eventually give the manuscript a more identifiable discussion section?

This reviewer would like to thank the authors in advance for their responses to this review, wishes the authors all success.

Round 2

Reviewer 1 Report

I still have doubts and the authors were unable to provide sound arguments to my concerns.

I agree with them that high-speed rail contributes to growth and innovation development. Innovation may occur either at the stage of infrastructure and trains construction and transportation service operation. However, two different economic models of development are implied. In the first case, innovation occurs in the place where R&D and manufacturing facilities are located. Agglomeration effects may also be relevant in this case. And the development of green technology innovations may also have its weight. I agree with authors when they affirm:

On the other hand, high-speed rail is a key element affecting the ecological environment and green growth of the region (Sun et al., 2020; Jia et al., 2021; Kong et al., 2021; Chen et al., 2021). Technological innovation produced by high-speed rail construction has environmental benefits, which can improve resource utilization efficiency, reduce pollution emissions to a greater extent, and reduce the pollution of the environment (Yang et al., 2019; Jia et al., 2021).

However, green technology innovations may also be generated by other industries, such as energy, automotive, electronic, ICT. Henceforth, authors need to distinguish the source of green technology innovation to investigate the linkage between innovation and high-speed rail.

Is this the economic model that the authors consider in their paper? In this case, I do not understand the reason they include the arrival/departure frequency of trains. The accumulation of knowledge needs several years to generate useful results. It is not a matter of minutes or hours. Time scale for innovation is years, sometimes months, not minutes or hours. So, why using a departure/arrival frequency variable to test for lag effects?

In the second case, innovation may occur because high rail transportation supports the movement of knowledge and competences embodied in high qualified people. Thus, as an effect, high speed rail supports the concentration of knowledge necessary to innovate. Is this the economic model underlining the proposed research? Why there should be a specific relationship between green technology innovation and people movement promoted by high-speed rail?

Authors should better explain what they want to investigate.

The authors should perform the Hausman test to choose between fixed effects model or a random effects model. The null hypothesis is that the preferred model is a random effects model. The alternate hypothesis is that the model that should be used is a fixed effects model. Such test looks to see if there is a correlation between the unique errors and the regressors in the model. The null hypothesis is that there is no correlation between the two. See, for instance

Amini, S., Delgado, M.S., Henderson, D.J. and Parmeter, C.F. (2012), "Fixed vs Random: The Hausman Test Four Decades Later", Baltagi, B.H., Carter Hill, R., Newey, W.K. and White, H.L. (Ed.) Essays in Honor of Jerry Hausman (Advances in Econometrics, Vol. 29), Emerald Group Publishing Limited, Bingley, pp. 479-513.
